# Thermal Evaporation Synthesis of Vertically Aligned Zn_2_SnO_4_/ZnO Radial Heterostructured Nanowires Array

**DOI:** 10.3390/nano11061500

**Published:** 2021-06-06

**Authors:** Gillsang Han, Minje Kang, Yoojae Jeong, Sangwook Lee, Insun Cho

**Affiliations:** 1School of Advanced Materials Science and Engineering, Sungkyunkwan University, Suwon 16419, Korea; hangillsang1@gmail.com; 2Department of Materials Science and Engineering, Ajou University, Suwon 16499, Korea; alswp965@ajou.ac.kr (M.K.); dox123@ajou.ac.kr (Y.J.); 3Department of Energy Systems Research, Ajou University, Suwon 16499, Korea; 4School of Materials Science and Engineering, Kyungpook National University, Daegu 41566, Korea

**Keywords:** thermal evaporation synthesis, Zn_2_SnO_4_/ZnO, heterostructured nanowires array, interface, charge transport

## Abstract

The construction of a heterostructured nanowires array allows the simultaneous manipulation of the interfacial, surface, charge transport, and transfer properties, offering new opportunities to achieve multi-functionality for various applications. Herein, we developed facile thermal evaporation and post-annealing method to synthesize ternary-Zn_2_SnO_4_/binary-ZnO radially heterostructured nanowires array (HNA). Vertically aligned ZnO nanowires array (3.5 μm in length) were grown on a ZnO-nanoparticle-seeded, fluorine-doped tin oxide substrate by a hydrothermal method. Subsequently, the amorphous layer consisting of Zn-Sn-O complex was uniformly deposited on the surface of the ZnO nanowires via the thermal evaporation of the Zn and Sn powder mixture in vacuum, followed by post-annealing at 550 °C in air to oxidize and crystallize the Zn_2_SnO_4_ shell layer. The use of a powder mixture composed of elemental Zn and Sn (rather than oxides and carbon mixture) as an evaporation source ensures high vapor pressure at a low temperature (e.g., 700 °C) during thermal evaporation. The morphology, microstructure, and charge-transport properties of the Zn_2_SnO_4_/ZnO HNA were investigated by scanning electron microscopy, X-ray diffraction, Raman spectroscopy, transmission electron microscopy, and electrochemical impedance spectroscopy. Notably, the optimally synthesized Zn_2_SnO_4_/ZnO HNA shows an intimate interface, high surface roughness, and superior charge-separation and -transport properties compared with the pristine ZnO nanowires array.

## 1. Introduction

Semiconductor metal oxide nanowires composed of earth-abundant elements are technologically essential materials for energy conversion/storage devices, optoelectronics, and sensors [1,2,3,4,5,6]. Nanowires often outperform the commonly utilized isotropic polycrystalline or particulate films in applications that require more complex and multifunctional materials [7,8,9]. This is because nanowires have two different-length scales (small diameter and significant length) that can be independently tailored to match the characteristic lengths of disparate physical processes. In addition, nanowires can also serve as building blocks for constructing heterostructured nanowires with designed materials that manipulate the surface, interface, and charge-transport/transfer properties, enabling multi-functionality [10,11]. For example, core/shell nanowires with type-II band alignment (staggered band edge alignment) spatially improve charge separation, leading to increased charge carrier lifetime and advantages in photocatalytic and photovoltaic performances [12,13]. 

Binary metal oxides, such as ZnO [8,14], TiO_2_ [7,15], and SnO_2_ [9,16], are the most widely used materials for electron transport in energy conversion applications and optoelectronic devices (e.g., photodiodes, dye-sensitized, and perovskite solar cells). Ternary metal oxides, such as Zn_2_SnO_4_ [17,18], BaSnO_3_ [19], and BaTiO_3_ [20], have also been investigated as alternatives, demonstrating improved performance and functionality for binary oxides. Recently, n-type semiconductor Zn_2_SnO_4_ (ZSO, zinc stannate) has attracted attention for its potential as a transparent conducting electrode, gas sensor, and perovskite solar cell owing to its bandgap energy of 3.6 eV, high mobility and conductivity, and low refractive index [21,22]. Thus far, diverse synthesis methods, including RF sputtering [23], pulsed laser deposition [24], hydrothermal method [25,26,27,28], vapor transport [22,23], and sol-gel spin-coating [29], have been explored to synthesize ZSO nanowires and their heterostructures with various morphologies. For example, Wang et al. synthesized ribbon-like ZSO nanowires through the vapor transport method at 800 °C without catalysts [30]. The synthesized nanowires exhibited an average width of 100–200 nm and ultra-long length of up to 1 mm. Mali et al. synthesized porous ZSO nanofibers by an electrospinning method that is used for perovskite solar cells [31]. Additionally, Bora et al. [32] and Siwatch et al. [33] reported a hydrothermal method to synthesize a ZSO/ZnO heterostructured nanowires array that exhibited improved functionality and photovoltaic performance. However, most previous studies have reported randomly aligned nanowires with often less controllability and uniformity in the nanowires or heterostructure morphology. 

In this study, we report a thermal evaporation method to synthesize vertically aligned ZSO/ZnO heterostructured nanowires array (HNA), demonstrating a highly aligned and uniform morphology. Single-crystalline ZnO nanowires array were first grown on the fluorine-doped tin oxide (FTO) substrate. Subsequently, the ZSO shell layer was formed by the thermal evaporation of the Zn and Sn metal mixture, followed by post-annealing at 550 °C. The thermal evaporation of the elemental metal mixture allows the control of the amount of Zn and Sn independently and ensures a high vapor pressure at a low temperature (700 °C). Notably, a highly crystalline ZSO shell layer with an average thickness of ~15 nm was successfully formed on the ZnO nanowires array. The resulting ZSO/ZnO HNA exhibited a higher surface roughness, close interface, and superior charge-transport properties than the pristine ZnO nanowires array.

## 2. Experimental Section

### 2.1. Deposition of ZnO Seed Layer

A ZnO nanoparticle seed layer was deposited by a sol-gel spin-coating method. The coating solution was prepared by dissolving zinc acetate dihydrate (0.263 g, ZnAc-2H_2_O, 99.9%, Sigma Aldrich Chemicals, St. Louis, USA) in anhydrous ethanol (20 mL, 99.9%, Daejung Chemicals). Acetylacetone (0.2 mL, 99.5%, Sigma-Aldrich Chemicals) was added as a stabilizer. After aging at 25 °C for 24 h, the resulting solution was spin-coated on pre-cleaned FTO substrates (TEC8, Pilkington) at 2500 rpm for 1 min. Subsequently, the samples were annealed at 350 °C for 1 h in air to form the ZnO nanoparticle seed layer and remove organic residues.

### 2.2. Hydrothermal Growth of ZnO Nanowires Array

ZnO nanowires array were grown on ZnO nanoparticle-seeded FTO substrates through a hydrothermal route. The growth solution was prepared by dissolving zinc nitrate hexahydrate (1.487 g, >99%, Sigma-Aldrich Chemicals) and hexamethylenetetramine (0.701 g, >99%, Sigma-Aldrich Chemicals) in deionized water (100 mL). After stirring for 10 min, polyethyleneimine (1.0 g, PEI, branched, *M*_w_ ~25,000, Sigma-Aldrich Chemicals) and ammonia (3.0 cc, 25–30%, Ducksan Chemicals, Seoul, Korea) were added and stirred for an additional 10 min. The growth solution was poured into a glass bottle (Schott bottle, 125 mL capacity). Then, the ZnO-seeded FTO substrates were vertically suspended in the solution. Finally, the growth solution was heated to 100 °C in an oven and held for 2–6 h. The obtained samples were washed with deionized water, followed by absolute ethanol, and dried with N_2_ in air. 

### 2.3. Synthesis of ZSO/ZnO Heterostructure Nanowires Array

The ZSO shell layer was synthesized by the thermal evaporation of an elemental Zn and Sn powder mixture in a tube furnace, followed by post-annealing in a muffle furnace. First, a Zn-Sn-O amorphous shell layer was deposited on the ZnO nanowires array (sample size: 2 cm × 2 cm) by the thermal evaporation of Zn/Sn (molar ratio of Zn/Sn = 2 and loading amount = 2 g) in a vacuum (1 mTorr, 700 °C for 0.5–2 h), followed by oxidation with O_2_ gas flowing (50 sccm). The substrate was positioned 10 cm away from the precursor crucible on the downstream side. Next, the samples were annealed at 550 °C for 1 h in an air atmosphere to form a crystalline ZSO shell layer on the surface of the ZnO nanowires array. 

### 2.4. Characterization and Measurement of Materials

The crystal structures of the synthesized materials were determined using X-ray diffraction (XRD, Mac-Science, M18XHF-SRA). The morphologies and film thicknesses were observed through field-emission scanning electron microscopy (JEOL, JSM-6330F). Transmission electron microscopy (TEM) images and selected area diffraction (SAD) patterns were recorded on a JEM-3000F (JEOL) microscope at an accelerating voltage of 300 kV. Raman spectra were recorded using a Raman spectrometer (Horiaba Jobin Yvon, T64000). Electrochemical impedance spectroscopy (EIS) measurements were conducted in sandwich-type cells with N719 dye, AN-50 electrolyte, Pt counter electrode, and working electrode under simulated sunlight illumination (AM 1.5 G, 100 mW/cm^2^). The amplitude of the sinusoidal voltage was 10 mV, and the examined frequency range was 7 MHz to 1 Hz. Mott–Schottky plots were measured using a three-electrode system (a Pt wire counter electrode and saturated calomel reference electrode) in the frequency range of 300–3000 Hz. 

## 3. Results and Discussion

Figure 1 shows a scheme of the synthesis process of ZSO/ZnO radial HNA on the FTO substrates. First, the ZnO nanoparticle film was deposited by sol-gel spin-coating (2500 rpm, 1 cycle) of the ZnO precursor solution (0.06 M), followed by annealing at 350 °C for 30 min to form a ZnO seed layer on the FTO substrate. The ZnO nanowire array (NW) was then grown on the ZnO-seeded FTO substrate using a hydrothermal method at 100 °C for 2 h. Subsequently, the ZnO NW were treated with vapors of the Zn and Sn mixture evaporated thermally at 700 °C in a vacuum (1 mTorr). Finally, they were annealed at 550 °C in air to form a crystalline ZSO shell layer on the ZnO NW. Notably, the thickness and morphology of the ZSO shell layer could be controlled by adjusting the thermal evaporation time.

Figure 2 shows the SEM images of the synthesized ZnO NW and ZSO/ZnO HNA. The growth conditions of the hydrothermal process (e.g., NH_4_OH amount, growth time, and cycle) were optimized to obtain dense and vertical ZnO NW on the FTO substrate (see Appendix A). The resulting ZnO NW exhibited a high-density and vertically aligned nanowire morphology, with an average length of approximately 3.5 μm (Figure 2a). In addition, the nanowires exhibited close contact with the FTO substrate. Notably, the nanowires had a smooth surface and tapered morphology near the tip (Figure 2b). As shown in Figure 2c, the ZSO/ZnO HNA also exhibited a comparable length of 3.6 μm. However, the ZSO/ZnO HNA exhibited a slightly larger nanowire diameter than the ZnO NW. Interestingly, their surface was much rougher than the ZnO NW because of the formation of nanoparticles at the surface (Figure 2d). 

XRD and Raman spectroscopy were performed to confirm the formation of the ZSO shell layer on the ZnO NW (Figure 3). Figure 3a shows the XRD patterns of the ZnO NW and ZSO/ZnO HNA. The ZnO NW exhibits a strong (002) peak intensity, indicating a preferred growth orientation along the [00l] direction. The ZSO/ZnO HNA also exhibits a high (002) peak intensity, retaining the [00l] preferred orientation of the ZnO NW. In addition, three additional weak peaks are observed at 17.6, 29.2, and 34.3°, which are indexed to the (111), (220), and (311) planes of the cubic Zn_2_SnO_4_ phase, respectively [32]. Figure 3b shows the Raman spectra of both ZnO NW and ZSO/ZnO HNA. The ZnO NW exhibits a broad peak centered at 443 cm^−1^, which corresponds to the E_2_ mode for ZnO [34]. After deposition of the ZSO shell layer, that is, for the ZSO/ZnO HNA, two peaks, at 443 and 673 cm^−1^, are observed, corresponding to the E2 mode for the ZnO and A_1g_ modes (stretching vibration mode of SnO_6_ octahedra) of spinel-type Zn_2_SnO_4_ [35]. Consequently, a ZSO shell layer with nanoparticle morphology was successfully formed on the ZnO NWs via thermal evaporation, followed by post-annealing.

TEM analyses were conducted to investigate the microstructure and interface of both ZnO NW and ZSO/ZnO HNA (Figure 4). Figure 4a,b show the TEM and high-resolution TEM images of the ZnO NW, respectively. The ZnO NW exhibits tapered tips and a smooth surface. The lattice fringes of 0.521 and 0.281 nm correspond to the (001) and (100) planes of hexagonal ZnO, respectively. In addition, the fast Fourier transform and selected area electron diffraction (SAED) patterns (Figure 4c) indicate that the ZnO NW has high crystallinity and a preferred growth direction of [1]. 

Figure 4d,e shows the TEM images of the ZSO/ZnO HNA. A thin nanoparticle layer covers the surface of the ZnO NW, consistent with the SEM observation (see inset of Figure 2d). They have an average thickness of ~15 nm (inset of Figure 4e). A high-resolution TEM image of the ZSO/ZnO HNA at the interface region is shown in Figure 4f. Notably, a highly crystalline layer with intimate contact with the ZnO NWs was formed. The lattice fringes of 0.433 and 0.310 nm were indexed to the (002) and (220) planes of cubic Zn_2_SnO_4_. TEM energy-dispersive spectroscopy indicated that the shell layer consisted of both Zn and Sn (Appendix A). Therefore, it is concluded that a crystalline ZSO/ZnO HNA with a high surface roughness and close interface was successfully synthesized on the FTO substrate by thermal evaporation and subsequent post-annealing. 

We tested different synthesis conditions, that is, (1) 30 min evaporation without post-annealing, (2) 30 min evaporation with post-annealing at 550 °C for 1 h, (3) 2 h evaporation with post-annealing at 550 °C for 1 h. As shown in Appendix A, an amorphous-like layer (low crystallinity) was formed without post-annealing. For the sample prepared with an evaporation time of 30 min and post-annealing at 550 °C/1 h, a much thinner crystalline layer (~6 nm) with less surface roughness was formed. When the evaporation time was increased to 2 h after post-annealing at 550 °C/1 h, a much thicker nanoparticle shell layer (~40 nm) was synthesized. Accordingly, the thickness of the ZSO shell layer could be controlled by adjusting the thermal evaporation time.

The conduction type, carrier concentration, and flat-band potential (*V_fb_*) values of ZnO NW and ZSO/ZnO HNA were determined from the Mott–Schottky measurements. The following equation describes the straight line in the Mott–Schottky curves, [36]
$$C_{sc}^{-2}=\frac{2}{q\varepsilon \varepsilon_{o}N_{D}}\left(V-V_{fb}-\frac{kT}{q}\right)$$
where $C_{sc}$ is the space charge capacitance, *q* is the elementary charge (1.602 × 10^−19^ C), $N_{D}$ is the carrier density, $\varepsilon_{0}$ is the vacuum permittivity, $\varepsilon_{r}$ is the dielectric constant, *V* is the applied potential, $V_{fb}$ is the flat-band potential, k is the Boltzmann constant, and T is the temperature. The Mott–Schottky plots and corresponding linear fits of the ZnO NW and ZSO/ZnO HNA are shown in Figure 5a. Both exhibit positive slopes, indicating that both electrodes are n-type semiconductors with electrons as the majority carriers. The calculated donor concentration (*N_D_*), which is inversely proportional to the straight-line slope of the ZnO NW and ZSO/ZnO HNA was 5.6 × 10^19^ and 2.6 × 10^19^ cm^−3^, respectively. The ZSO/ZnO HNA exhibited N_D_ two times smaller than the ZnO NW, ascribed to the ZSO shell layer. In general, ZnO exhibited a number of intrinsic defects, and thus it exhibited a high charge carrier density even without additional treatments [37]. On the other hand, the ZSO exhibited smaller donor concentration, which is attributed to the post annealing that reduces the electron donors such as oxygen vacancies. 

The *V_fb_* is the electrochemical potential value at which the band bending disappears. It is close to the conduction band edge position in the case of doped n-type semiconductor. Notably, the *V_fb_* of the ZnO NW and ZSO/ZnO HNA were −0.02 and −0.24 V vs. RHE, respectively. The conduction and valence band-edge positions of ZnO and ZSO/ZnO NWs were determined from the bandgap and *V_fb_* values by assuming the difference between the conduction band-edge and the *V_fb_* to be insignificant. The ZSO/ZnO HNA exhibited a significantly negative *V_fb_* value, indicating that its conduction band edge was higher than that of the ZnO NW. Figure 5b shows the estimated energy band edge positions of the ZnO NW and ZSO/ZnO HNA. According to the results, both the conduction and valence band edges of ZSO were positioned above those of the ZnO NW. Accordingly, the ZnO and ZSO heterostructures had a staggered band edge; that is, they form a type-II heterojunction, which improves the spatial charge separation [32,38]. 

The charge-transport properties of both electrodes (ZnO NW and ZSO/ZnO HNA) were evaluated by EIS measurements [39]. As shown in Figure 5c, the ZSO/ZnO HNA exhibited a smaller semicircle than the ZnO NW, indicating reduced charge transport and transfer resistance values [40]. In addition, the relative surface area was estimated using a dye-adsorption method (Appendix A), suggesting that the ZSO/ZnO HNA has a 130% larger surface area than the ZnO NW. 

As a result, the construction of the ZSO/ZnO HNA improved charge separation, transport, and transfer (injection) properties (Figure 5d), which is attributed to the formation of type-II heterojunctions, intimate interfaces, and superior surface roughness compared to the ZnO NWs.

## 4. Conclusions

We successfully synthesized a ZSO/ZnO HNA via facile thermal evaporation and post-annealing method. First, the ZnO nanowires array was grown on a ZnO nanoparticle-seeded FTO substrate through a hydrothermal method. Then, an amorphous shell layer composed of Zn-Sn-O was uniformly formed on the ZnO nanowire surface by the thermal evaporation of the Zn and Sn mixture at 700 °C in vacuum, followed by post-annealing at 550 °C in air to synthesize the crystalline ZSO shell layer. XRD and Raman analyses confirmed the formation of the ZSO shell layer on the ZnO nanowires array. Interestingly, the SEM and TEM analyses revealed that the deposited ZSO exhibits a highly crystalline nanoparticle morphology and is closely in contact with the surface of the ZnO nanowires without any voids. The optimally synthesized ZSO/ZnO HNA showed a higher surface roughness and superior charge-separation/transport properties as compared with the ZnO nanowires array. With further optimization of the ZSO layer (e.g., thickness), our ZSO/ZnO HNA can be applied as an electron-transporting layer to various energy-conversion devices, such as dye/quantum-dot sensitized devices, perovskite solar cells, and photoelectrochemical cells.

## Figures and Tables

**Figure 1 nanomaterials-11-01500-f001:**
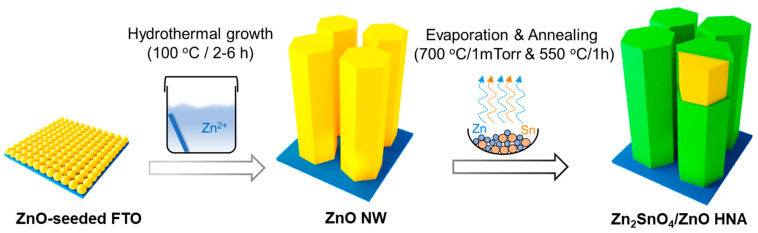
Synthesis of the Zn_2_SnO_4_/ZnO heterostructured nanowires array (HNA). Step 1. Sol-gel spin-coating of the ZnO seed layer on the FTO substrate. Step 2. Hydrothermal growth of ZnO NWs at 100 °C for 2 h. Step 3. Thermal evaporation of Zn and Sn mixture at 700 °C for 1 h in vacuum (1 mTorr), followed by post-annealing at 550 °C for 1 h.

**Figure 2 nanomaterials-11-01500-f002:**
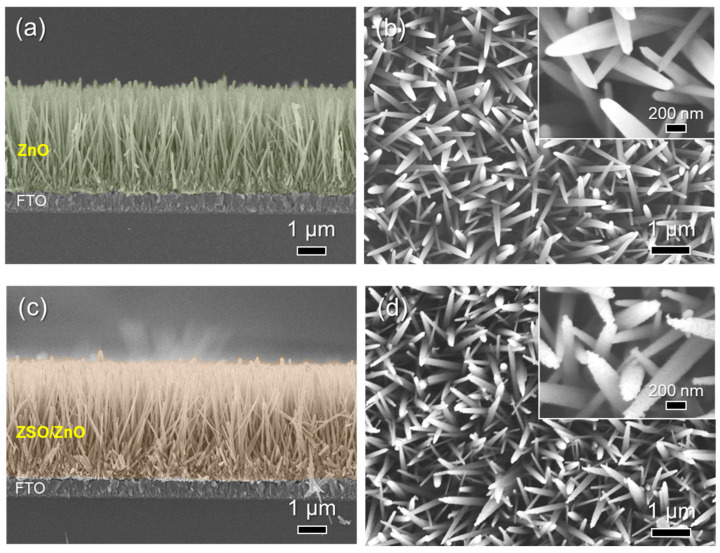
Morphological characterization. Cross- and top-view SEM images of (**a**,**b**) ZnO NW and (**c**,**d**) ZSO/ZnO HNA. The ZSO/ZnO HNA exhibited rough and lumpy surface.

**Figure 3 nanomaterials-11-01500-f003:**
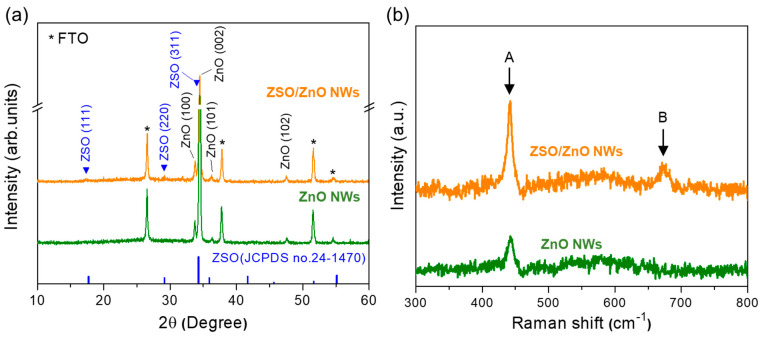
X−ray diffraction (XRD) and Raman spectroscopy of ZnO and ZSO/ZnO HNA. (**a**) XRD patterns. (**b**) Raman spectra. A: E_2_ mode for ZnO. B: Stretching vibration mode of SnO_6_ octahedra in Zn_2_SnO_4_.

**Figure 4 nanomaterials-11-01500-f004:**
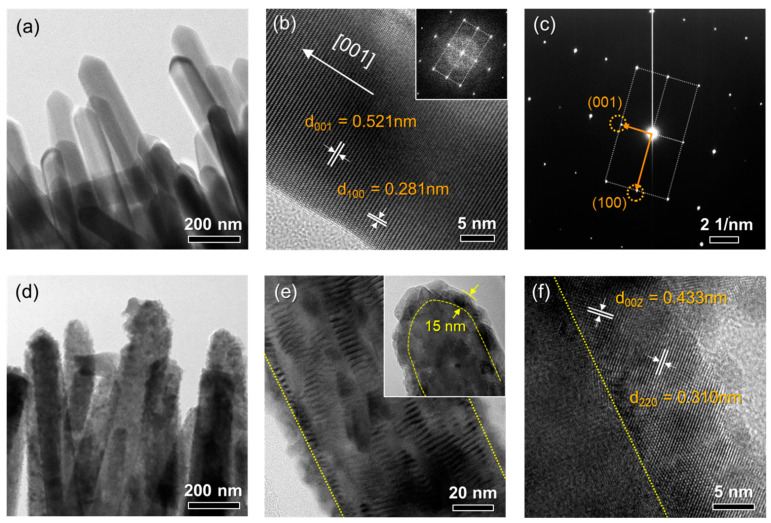
Transmission electron microscopy (TEM). (**a**) TEM, (**b**) high-resolution TEM images and (**c**) selected area diffraction pattern of ZnO NW. (**d**,**e**) TEM and (**f**) high-resolution TEM images of ZSO/ZnO HNA.

**Figure 5 nanomaterials-11-01500-f005:**
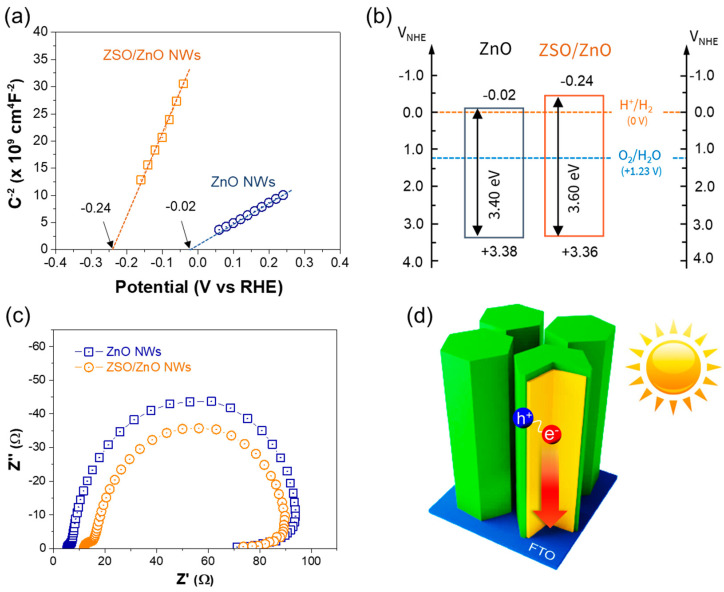
(**a**) Mott−Schottky analysis. (**b**) Energy band position of ZnO NW and ZSO/ZnO HNA. (**c**) Electrochemical impedance spectroscopy. (**d**) Scheme of the enhanced charge separation and transport in the ZSO/ZnO HNA.

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
