# Peer review of "Thermal Evaporation Synthesis of Vertically Aligned Zn_2_SnO_4_/ZnO Radial Heterostructured Nanowires Array"

_nanomaterials, 2021, doi:10.3390/nano11061500_

Round 1
Reviewer 1 Report
The manuscript titled "Thermal Evaporation Synthesis of Vertically Aligned Zn2SnO4 / ZnO Radial Heterostructured Nanowire Arrays" shows interesting research results. However, nothing else is included, there is no discussion of the results!
The supplement contains the results of studies on the influence of various factors on the growth of ZnO nanowires, but nowhere is any comment given as to why these factors cause such effects. The research results presented in Figure S5 are very interesting and it is necessary to explain them.
Figure S4 does not fully confirm the authors' suggestion that the Zn2SnO4 compound is formed on the surface of ZnO nanowires.
Figure S6 is unnecessary, because Figs. 2 and 4 show that the specific surface of the nanowires with a layer deposited on their surface has a greater development.
Additional remarks:
- The “molar ratio of Zn / Sn = 2,2 g” is not reported in units of mass.
- "In addition, the nanowires exhibit intimate contact with the FTO substrate without voids." - This cannot be clearly stated on the basis of Figure 2d.
- "Interestingly, their surface was much rougher than the ZnO NW because of the formation of nanoparticles at the surface (Figure 2d)" why nanoparticles are formed when Zn and Sn evaporate and separate as pairs.
- TEM investigation - How were samples prepared for TEM observation?
- "According to the results, both the conduction and valence band edges of ZSO are positioned above those of the ZnO NW." - Why?
6. “As shown in Figure 5c, the ZSO/ nO HNA exhibited a smaller semicircle than the ZnO NW, indicating reduced charge transport and transfer resistance values” - Why? The resistance is proportional to the concentration of the electric charge carriers. Therefore, if the concentration of carriers is higher, the resistance is lower and the semicircle of the ZnO NW should be lower!
Reviewer 2 Report
The authors report on the fabrication and properties of ZnO/ZnSnO nanowires. These are produced by coating ZnO nanowires with Zn and S with a post-oxidation treatment in air.
The paper is clearly written and Figures support the text and are nicely arranged. The SEM, TEM, XRD and Raman results are ok. As to Fig. 5, I’m not expert enough to judge the details.
I recommend publication, but the authors have to discuss the following queries:
- In Ref.33 similar structures were produced. Please mention in detail in which respect your data are new regarding growth and properties.
- Page 2: “…functionality compared to binary oxides”?
- In the synthesis part of ZnSnO you mention, that a Zn-Sn-O amorphous layer was deposited in vacuum. But you mention that only Zn and Sn were evaporated. Where does the oxygen come from? I suspect that you first deposited ZnSn and oxidized it afterwards, right?
- Figure 1 should be shown in the Experimental section.
- The first paragraph of the “Results and Discussion” is a repetition of the Experimental part. Please delete this.
- 3a: The ZSO peak at 34.4° is not visible. Either enlarge the plot to show this.
- 22,33,35: Please correct the “doi”-link.
- 7,13: Delete “”.
- 19: “(80-.)” ?
Reviewer 3 Report
The manuscript reads more like a report of observations than a scientific paper. I would have liked to see more discussions and conclusions, rather than just results.
The seed-layer needs to be defined better. It is not clear if it is a solid mono crystallin film, or a layer of densely packed and randomly oriented nano-sized particles?
Depending on the nature of the seed layer, I would like to see a discussion of what determines the growth direction. The wires grow in the (001) direction, yet they are mainly vertical.
The EIS and Mott-Schottky techniques need to be introduced and referenced as they are not commonly used techniques and the interpretation of the results are not clear to me.
The wires are described as tapered and 3.5 um in length. Any number for the average diameter? Any number on the spread of both numbers? The tapering is visible in the top view SEM images, but not in the TEM images. In the TEM images, the tip is faceted, but the tapering is not clear. Are we presented with different diameters in the images? The SEM wires seem to be significantly wider.
There appears to be a variation in the tilt of the wires. How does that influence the XRD data? the presentation of the XRD data is not very good. Perhaps it would be clearer in log-scale? The spectra are indexed with stars (the FTO substrate) and triangles (?) without any explanation. there is also something at the bottom of the spectra, ZSO.... Explain!
The ZSO "layer" is described as having an average thickness of 15nm. How is this defined, as the "layer" appears to be a coating of nano-particles?
In the last paragraph on page 8, the authors describe three conditions referred to as (1), (2) and (c). Why the "c"? The annealing is described as 550°C / 1h. Does that mean for 1h at 550°C? This needs to be written a bit better.
Where does the equation on page 9 come from? The measurements should be described a bit better and definitely referenced.
What is the significance of the flat-band voltage. There is a hint that it gives the top of the valence band, but it should be discussed in the manuscript. It is stated that this gives the alignment of the conduction bands, without any explanation. It this from knowledge of the bandgaps?
From the slope of the plot, the donor concentration can be determined. I can understand this for the naked wires, but how does the coating influence the measured value? what is the significance of the difference the donor concentration? This must be discussed!
The last sentence of the first paragraph on page 10 states that the structures have "superior surface roughness". What does that mean, as all images show a very rough surface, consisting of a particle-coated surface?
One intriguing part of the SI is that the length of the wires do not depend on the growth time, only on the amount of NH4OH. The reason from this could be discussed in the manuscript.
Reviewer 4 Report
The manuscript "Thermal Evaporation Synthesis of Vertically Aligned Zn2SnO4/ZnO Radial Heterostructured Nanowire Arrays" presents the results on the synthesis and characterization of Zn2SnO4/ZnO nanowires.
The methodology is appropriate and described in enough details, the results are well explained and support the conclusions.
I recommend its publication with minor changes.
I suggest the following changes (do not affecting the overall scientific and technical quality):
-In experimental section please include the FTO provider or if it is synthesized in lab please state it.
-In Results and Discussion please change “Fig 1 shows the synthesis process” by “Fig 1 shows a scheme/diagram of the synthesis process”
-In Conclusion section the last phrase suggests that “With further optimization in length, diameter and morphology… “ please be more explicit. E.g. decreased/increased length, diameter and which morphology.
From Figure S6. Authors conclude a 130% of increased surface area. Please support the figure with an extended explanation or analysis because the interpretation may be not direct. Please add also ticks to the vertical axis.
Reviewer 5 Report
The manuscript of Cho and coauthors deals with the synthesis of a nanocomposite heterostructure based on ZnO/Zn2SnO4 for photocatalytic applications.
The manuscript is well organized, the whole characterization very impressive and the results quite interesting.
In my opinion this manuscript deserves to be published in the present form.
Round 2
Reviewer 1 Report
Authors did not take into account the comments made in the first review.
Additionally, please explain:
- the statements "Single-crystalline ZnO nanowire arrays were first grown on the fluorine-doped tin oxide (FTO) substrate." - on what basis do the authors claim to have received single-crystalline nanowire?
- The statements "First, a Zn-Sn-O amorphous shell layer was deposited on the ZnO nanowire array (sample size: 2 cm × 2 cm) by the thermal evaporation ..." - on what basis the authors claim that an amorphous layer is formed. Under such conditions, metals do not crystallize in amorphous form.
- TEM investigation - How were samples prepared for TEM observation?
- “As shown in Figure S5, an amorphous layer was formed without post-annealing” - this cannot be stated based on this figure.
- During the Mott – Schottky measurements what V applied?
- "According to the results, both the conduction and valence band edges of ZSO are positioned above those of the ZnO NW." - Why?
- The size of the semicircle (Figure 5c) is not only a consequence of the change in resistance and charge transfer capacity. The analysis should take into account the FTO, electrolyte and the bulk semiconductor resistances with the barrier capacitance. In addition, blocking properties and presence of surface states should be taken into account.
- “In addition, the relative surface area was estimated using a dye-adsorption method (Figure S6), suggesting that the ZSO/ZnO HNA has a 130% larger surface area than the ZnO NW. Therefore, the construction of the ZSO/ZnO HNA improved charge separation, transport, and transfer (injection) properties (Figure 5d), which is attributed to the formation of type-II heterojunctions, intimate interfaces, and superior surface roughness comparing to the ZnO NWs." - the increase in surface area cannot be explained by everything, i.e. improved charge separation, transport, and transfer (injection) properties. What is the size of the specific surface area for the formation of type-II heterojunctions? On what basis do the authors claim that a type II heterojunction is formed?
Author Response
Response to Reviewer 1 Comments (Round 2)
Comment 1: Authors did not take into account the comments made in the first review.
Response 1: We do not agree with the reviewer. There might be some technical issue/error in the MDPI system. We had taken into account all the comments raised by reviewer 1 and answered all the questions/comments before (1st round revision, see below).
*************************************************************************
(Round 1 point-by-point response by the authors)
Comment 1: The manuscript titled "Thermal Evaporation Synthesis of Vertically Aligned Zn2SnO4 / ZnO Radial Heterostructured Nanowire Arrays" shows interesting research results. However, nothing else is included, there is no discussion of the results!
Response 1: We appreciate the reviewer's suggestion.
As the reviewer pointed out, we did our best to add more discussions. All the changes are marked with 'red color' in the revised manuscript.
Comment 2: The supplement contains the results of studies on the influence of various factors on the growth of ZnO nanowires, but nowhere is any comment given as to why these factors cause such effects. The research results presented in Figure S5 are very interesting and it is necessary to explain them.
Response & Revision 2: We agree with the reviewer.
As requested, more discussions on Figure S1 – S3 was added in the revised supplementary material as follows:
(Figure S1) The addition of NH4OH affected the concentration of Zn-complex that largely affects the Zn solubility in the growth solution. Consequently, the supersaturation, i.e., nuclei density, can be controlled. Additionally, the pH affected growth rate [Nanoscale Research Letters 13 (2018) 249; Inorg. Chem. 45 (2006) 7535–7543]. Therefore, the aspect ratio and density of ZnO NWs increased with the addition of NH4OH.
(Figure S2) The growth time has little impact on the morphology and length of ZnO NWs. This result indicates that the growth rate is fast and whole growth occurs within 2 h. After 2 h growth, most of the precursors are consumed, i.e., no additional growth occurred.
(Figure S3) As repeating the growth cycles, the nanowire length increases linearly from 3.6 μm to 11.7 μm. There was no branch growth. However, NW density at the bottom increased.
Comment 3: Figure S4 does not fully confirm the authors' suggestion that the Zn2SnO4 compound is formed on the surface of ZnO nanowires.
Response 3: We agree with the reviewer.
As the reviewer pointed out, Figure S4 alone is not conclusive. However, from the XRD data shown in Figure 3a, the formation of crystalline Zn2SnO4 was confirmed. Therefore, our claim that the Zn2SnO4 compound is formed on the surface of ZnO nanowire is correct.
Comment 4: Figure S6 is unnecessary, because Figs. 2 and 4 show that the specific surface of the nanowires with a layer deposited on their surface has a greater development.
Response 4: We agree with the reviewer.
We intended to show the direct evidence of the enhanced surface area via an additional experiment. So we keep Figure S6 in the revised supplementary material.
Comment 5: Additional remarks:
Response 5: We revised as follows.
- The "molar ratio of Zn / Sn = 2,2 g" is not reported in units of mass.
--> (molar ratio of Zn/Sn=2 and the loading amount = 2g)
- "In addition, the nanowires exhibit intimate contact with the FTO substrate without voids." - This cannot be clearly stated on the basis of Figure 2d.
--> the nanowires exhibit intimate contact with the FTO substrate without voids.
- "Interestingly, their surface was much rougher than the ZnO NW because of the formation of nanoparticles at the surface (Figure 2d)" why nanoparticles are formed when Zn and Sn evaporate and separate as pairs.
--> As we already noted, a continuous amorphous layer was deposited on the surface ZnO NWs. After a post-annealing, the Zn2SnO4 with nanoparticle morphology was formed (See Figure S6).
- TEM investigation - How were samples prepared for TEM observation?
--> The nanowires are scratched from the substrate by using a razor, and then it dispersed in the IPA solution with ultrasonication, followed by drop-casting on the TEM grid.
- "According to the results, both the conduction and valence band edges of ZSO are positioned above those of the ZnO NW." - Why?
--> Mott-Schottky measurement gives the flat band potential value, and it is generally considered to be close to the conduction band edge (in the case of an n-type semiconductor). Based on the Mott-Schottky measurement, the flat band potential value of ZSO was higher (-0.24 V vs. RHE) than that of ZnO (-0.02 V vs. RHE). This result indicates that the conduction band edge of ZSO is located above that of ZnO. Next, based on the bandgap values, the valence band edge of ZSO is located above that of ZnO (See Figure 5b).
Comment 6: "As shown in Figure 5c, the ZSO/ ZnO HNA exhibited a smaller semicircle than the ZnO NW, indicating reduced charge transport and transfer resistance values" - Why? The resistance is proportional to the concentration of the electric charge carriers. Therefore, if the concentration of carriers is higher, the resistance is lower and the semicircle of the ZnO NW should be lower!
Response 6: We do appreciate the reviewer's comment.
As the reviewer pointed out, the conductivity (or resistivity) of the film is affected by three factors (charge carrier density, elemental charge, and mobility). Although the charge carrier density of ZSO/ZnO NWs is lower than the ZnO NWs, the electron mobility of ZSO (10-15 Cm2/V-sec) is higher than the ZnO (~5.5 cm2/V-sec). In addition, the EIS spectra were recorded under simulated sunlight illumination. So the photogenerated charge carriers and their dynamics (charge separation, transport, and transfer) strongly affect the EIS data. Therefore, the photogenerated carriers, type II heterojunction, and mobility are simultaneously affecting the enhanced charge transport and transfer properties.
******************************************************************************
Comment 2: Additionally, please explain: the statements "Single-crystalline ZnO nanowire arrays were first grown on the fluorine-doped tin oxide (FTO) substrate." - on what basis do the authors claim to have received single-crystalline nanowire?
Response 2: We do appreciate the reviewer’s comment.
It is well-known that the hydrothermal method we adopted allows growing well-defined single-crystalline ZnO nanowires [Nanotechnology 26, 355704 (2015); The Journal of Physical Chemistry C 114 (15), 7185-718]. We also checked the single-crystalline nature of our ZnO NWs by using TEM and SAED pattern analysis. As shown below, well-defined lattice images and spots are clearly seen. This result indicates that the ZnO NW is single-crystalline.
< High-resolution TEM image and corresponding SAED pattern>
Comment 3: The statements "First, a Zn-Sn-O amorphous shell layer was deposited on the ZnO nanowire array (sample size: 2 cm × 2 cm) by the thermal evaporation ..." - on what basis the authors claim that an amorphous layer is formed. Under such conditions, metals do not crystallize in amorphous form.
Response 3: We do appreciate the reviewer’s comment.
From the TEM observation (See Figure S5), we claimed the deposited Zn-Sn-O layer is amorphous without post-annealing. In our experiment, oxygen gas has flowed after the evaporation for oxidation. So the deposited layer is not a metallic phase.
Comment 4: TEM investigation - How were samples prepared for TEM observation?
Response 4: The nanowires are scratched from the substrate by using a razor, and then it dispersed in the IPA solution with ultrasonication, followed by drop-casting on the TEM grid (The same question is already answered in reviewer #1-comment 4, See the above).
Comment 5: “As shown in Figure S5, an amorphous layer was formed without post-annealing” - this cannot be stated based on this figure.
Response & Revision 5: We agree with the reviewer.
As the reviewer pointed out, the TEM image is not conclusive. So we changed the sentence as follows.
“As shown in Figure S5, an amorphous-like layer (low crystallinity) was formed without post-annealing.
Comment 6: During the Mott – Schottky measurements what V applied?
Response 6: As shown in Figure 5a, we applied from -0.4 V to +0.3 V vs. RHE to measure the Mott-Schottky plot (scan direction: anodic, i.e., from left to right side).
Comment 7: "According to the results, both the conduction and valence band edges of ZSO are positioned above those of the ZnO NW." - Why?
Response 7: Based on the Mott-Schottky analysis, the flat band potential (which is close to the conduction band edge for the n-type semiconductor) of ZnO NW and ZSO/ZnO HNA determined to be -0.24 and -0.02 V vs. RHE, respectively. This result indicates that the conduction band edge of ZSO/ZnO HNA is positioned above the conduction band edge of ZnO NW. Additionally, the valence band edges were determined from the bandgap value (i.e., by subtracting the band gap from the conduction band edge potential). Accordingly, the valence edge positions of ZnO NW and ZSO/ZnO HNA are +3.38 and +3.36 V vs. RHE, respectively. These results are already shown in Figure 5b.
Comment 8: The size of the semicircle (Figure 5c) is not only a consequence of the change in resistance and charge transfer capacity. The analysis should take into account the FTO, electrolyte and bulk semiconductor resistances with the barrier capacitance. In addition, blocking properties and the presence of surface states should be taken into account.
Response 8: We agree with the reviewer.
As the reviewer pointed out, all the parameters including FTO, electrolyte and bulk resistance should be considered for quantitative analysis of the EIS data. Our paper mostly focused on the synthesis of heterostructured nanowires and their advantages for energy applications (e.g., solar cell and photoelectrochemical water splitting). We just roughly analyzed the EIS data to give a brief and relative advantage for the charge transport and transfer properties. Therefore, we checked the last semicircle that contains the information on the charge transport (bulk resistance) and transfer properties for just comparison with the ZnO NW.
Comment 9: “In addition, the relative surface area was estimated using a dye-adsorption method (Figure S6), suggesting that the ZSO/ZnO HNA has a 130% larger surface area than the ZnO NW. Therefore, the construction of the ZSO/ZnO HNA improved charge separation, transport, and transfer (injection) properties (Figure 5d), which is attributed to the formation of type-II heterojunctions, intimate interfaces, and superior surface roughness comparing to the ZnO NWs." - the increase in surface area cannot be explained by everything, i.e. improved charge separation, transport, and transfer (injection) properties. What is the size of the specific surface area for the formation of type-II heterojunctions? On what basis do the authors claim that a type II heterojunction is formed?
Response 9: We agree with the reviewer.
There was a mistake in the writing. Except for the enlarged surface area, all the other properties (i.e., improved charge separation, transport and transfer) come from the EIS and Mott-Schottky analyses. For clarity, we changed the sentence as follows.
“The charge-transport properties of both electrodes (ZnO NW and ZSO/ZnO HNA) were evaluated by EIS measurements [39]. As shown in Figure 5c, the ZSO/ZnO HNA exhibited a smaller semicircle than the ZnO NW, indicating reduced charge transport and transfer resistance values [40]. In addition, the relative surface area was estimated using a dye-adsorption method (Figure S6), suggesting that the ZSO/ZnO HNA has a 130% larger surface area than the ZnO NW.
As a result, the construction of the ZSO/ZnO HNA improved charge separation, transport, and transfer (injection) properties (Figure 5d), which is attributed to the formation of type-II heterojunctions, intimate interfaces, and superior surface roughness compared to the ZnO NWs.”

Reviewer 3 Report
The revised version is OK for publication.
Author Response
We do appreciate the reviewer's positive comment.